# Interpolating Strange Attractors via Fractional Brownian Bridges

**DOI:** 10.3390/e24050718

**Published:** 2022-05-17

**Authors:** Sebastian Raubitzek, Thomas Neubauer, Jan Friedrich, Andreas Rauber

**Affiliations:** 1TU Wien, Information and Software Engineering Group, Favoritenstrasse 9-11/194, 1040 Vienna, Austria; thomas.neubauer@tuwien.ac.at (T.N.); rauber@ifs.tuwien.ac.at (A.R.); 2ForWind, Institute of Physics, University of Oldenburg, Küpkersweg 70, 26129 Oldenburg, Germany; jan.friedrich@uni-oldenburg.de

**Keywords:** time series interpolation, phase space reconstruction, Takens’ theorem, interpolation, stochastic interpolation, genetic algorithm, time series data, preprocessing, strange attractor, attractor, attractor reconstruction

## Abstract

We present a novel method for interpolating univariate time series data. The proposed method combines multi-point fractional Brownian bridges, a genetic algorithm, and Takens’ theorem for reconstructing a phase space from univariate time series data. The basic idea is to first generate a population of different stochastically-interpolated time series data, and secondly, to use a genetic algorithm to find the pieces in the population which generate the smoothest reconstructed phase space trajectory. A smooth trajectory curve is hereby found to have a low variance of second derivatives along the curve. For simplicity, we refer to the developed method as *PhaSpaSto*-interpolation, which is an abbreviation for **pha**se-**spa**ce-trajectory-smoothing **sto**chastic interpolation. The proposed approach is tested and validated with a univariate time series of the Lorenz system, five non-model data sets and compared to a cubic spline interpolation and a linear interpolation. We find that the criterion for smoothness guarantees low errors on known model and non-model data. Finally, we interpolate the discussed non-model data sets, and show the corresponding improved phase space portraits. The proposed method is useful for interpolating low-sampled time series data sets for, e.g., machine learning, regression analysis, or time series prediction approaches. Further, the results suggest that the variance of second derivatives along a given phase space trajectory is a valuable tool for phase space analysis of non-model time series data, and we expect it to be useful for future research.

## 1. Introduction

Many real-life time series data sets originate from complex systems and/or non-linear phenomena. Often these data sets are sparsely sampled as, e.g., long-term temperature, yield, or environmental data sets. The non-linear and stochastic nature of these data sets, in addition to being sparsely sampled, make predictions and analysis rather challenging. Thus, one tends to employ data augmentation or interpolation techniques.

Typical examples where such data augmentation techniques are deployed include gaps in time series from solar wind measurements [1,2], spatio-temporal wind fields from meteorological mast arrays [3], as well as the study of particle transport in intergalactic magnetic fields [4]. An enhancement of data in these examples is commonly achieved by relying on one of the various interpolation techniques, such as linear, polynomial, fractal, or stochastic interpolation methods. Nonetheless, choosing a suitable interpolation method can be difficult, and one should take into account the characteristics of the data at hand. For fluctuative and inherently random data, one would choose a stochastic interpolation as discussed in [5]. However, regarding deterministically chaotic systems where one can reconstruct a phase space based on Takens’ theorem [6], the choice is not so clear. Here, we present a method taking into account the properties of a data’s reconstructed phase space. Thereby, we want our reconstructed phase space trajectories to be as smooth as possible. This is achieved by combining multi-point Brownian bridges [5] and a genetic algorithm. For simplicity, we refer to the developed method as *PhaSpaSto*-interpolation, which is an abbreviation for **pha**se-**spa**ce-trajectory-smoothing **sto**chastic interpolation. We show, that the developed method performs well for the reconstructed phase space of the Lorenz system and several univariate, sparsely-sampled time series data. The results show that the presented method can effectively interpolate the Lorenz system and some of the discussed non-model data sets with comparatively low errors on known data points and convincing phase space portraits.

Furthermore, many of today’s most employed time series analysis and prediction techniques are from the domain of machine and/or deep learning. These methods are data-based, i.e., they learn from data; thus, a sufficient amount of data and data of good quality are necessary to, e.g., train a neural network. It is shown that interpolating time series data using a fractal or linear interpolation can improve the accuracy of the algorithm drastically [7]. We thus suggest PhaSpaSto-interpolation to be tested and used for data-based learning algorithms. Further, the presented criterion to guarantee a smooth phase space trajectory, i.e., the variance of second derivatives along a trajectory, is shown to be valuable for non-linear time series analysis. It can effectively identify interpolations with low errors for the presented model and non-model data sets. Thus, the presented *PhaSpaSto*-interpolation and the corresponding criterion should be considered when dealing with complex non-linear data sets.

This article is structured as follows: Section 2 collects publications related to the developed method and discusses them briefly. Section 3 describes the multi-point Brownian bridges [5], the Lorenz system, and the employed genetic algorithm, and further sums up the developed scheme. All results with the corresponding error tables and figures are discussed in Section 4. Section 5 concludes the findings of this article.

## 2. Related Work

The presented research is motivated by findings of [7,8]. It is further based on the stochastic interpolation method presented in [5]. Thus, we will briefly describe the mentioned publications, and list them chronologically, i.e., by their date of publication.

Ref. [9]: This publication presents a method to determine if images are blurry. For this purpose, the second derivatives of grey-scale images are calculated pixel per pixel, and the corresponding variance is analyzed for all pixels and their neighbors. If the variance is below a certain threshold, the image is blurry. This concept is used in the presented article. We adapted the idea of variances of second derivatives, which is discussed in Section 3.3.1.Ref. [10]: This research applies a combination of inverse distance methods, fuzzy set theory, and a genetic algorithm to interpolate rainfall data. The genetic algorithm was used to determine the parameters of the corresponding fuzzy membership functions. Thus, the idea of improving interpolation techniques is adapted from this publication.Ref. [5]: This publication presents a novel interpolation technique where the idea of a Brownian bridge, i.e., a constrained fractional Brownian motion (fBm), is extended to more than two points, i.e., to *multi-point fractional Brownian bridges*. The authors present an explicit construction that operates linearly on the fBm and can thus be interpreted as a Gaussian random process constrained on multiple, prescribed points. Further applications of this method are presented, such as determining optimal Hurst exponents for sparsely sampled time series filled up by multi-point fractional Brownian bridges with varying Hurst exponents. This method is used in the presented research to fuel the genetic algorithm.Ref. [7]: This publication presents a fractal interpolation to interpolate univariate time series data. The proposed method considers the Hurst exponent of the data under study. The authors show that fractal interpolation can increase the predictability of a given univariate time series. This research suggests that different interpolation methods for univariate time series data may yield predictions of different qualities. Thus, as presented here, an attractor-based interpolation is an obvious next step in contrast to a fluctuation-based interpolation.Ref. [8]: This publication is a continuation of [7]. The fractal interpolation and LSTM neural network approach are continued as ensembles of predictions. Randomly parameterized LSTM neural networks are generated from non-, linear-, and fractal-interpolated data. Afterward, these predictions are filtered based on their signal complexities. Some of the mentioned complexity measures require a suitable phase space embedding of the data under study and are related to the presented research in this article. Further, some of the data sets used here are discussed and predicted. We expect LSTM neural network predictions of stochastically interpolated data to outperform other interpolated approaches when considering the reconstructed phase space.Ref. [11] describes a multi-point reconstruction of a given time series. The method is based on the assumption of Markovianity of the time series. A refinement algorithm is presented, which allows to systematically fill up data points based on the empirically determined transition probability from one level to the next.

## 3. Methodology

The developed method consists of two steps. Firstly, generating a population of stochastically-interpolated time series data using multi-point fractional Brownian bridges; see Section 3.1. Secondly, these multi-point fractional Brownian bridges are improved via a genetic algorithm to minimize the variance of second-order derivatives along the reconstructed phase space trajectory (see Section 3.3). The whole scheme is depicted in Figure 1. Finally, we briefly discuss the Lorenz system and its implementation in Section 3.4.

### 3.1. Multi Point Fractional Brownian Bridges

As depicted in Figure 1, the employed genetic algorithm is fueled by a population of stochastically-interpolated time series data, in our case *multi-point fractional Brownian bridges*. To generate these stochastically-interpolated time series data, multi-point fractional Brownian bridges [5] were used. An in-depth discussion and corresponding applications of regular fractional Brownian bridges are provided in [12,13].

We consider a Gaussian random process X(t) whose covariance is defined as $C\left(t,t^{\prime }\right)=\langle X\left(t\right)X\left(t^{\prime }\right)\rangle$. In the following, we focus on fractional Brownian motion with the covariance given as $\langle X\left(t\right)X\left(t^{\prime }\right)\rangle =\frac{1}{2}\left(t^{2H}+t^{\prime 2H}-{\left|t-t^{\prime }\right|}^{2H}\right)$, where *H* is the Hurst exponent. To elucidate our interpolation scheme, we first define a so-called fractional Brownian bridge [14,15], which is a construction of fBm starting from 0 at t=0 and ending at $X_{1}$ at $t=t_{1}$, i.e.,
(1)$$X^{B}\left(t\right)=X\left(t\right)-\left(X\left(t_{1}\right)-X_{1}\right)\frac{\langle X\left(t\right)X\left(t_{1}\right)\rangle }{\langle X{\left(t_{1}\right)}^{2}\rangle }.$$
This construction ensures that $X^{B}\left(t_{1}\right)=X_{1}$. This single bridge can now be generalized to an arbitrary number of (non-equidistant) prescribed points $X_{i}$ at $t_{i}$ by virtue of a multi-point fractional Brownian bridge [5]
(2)$$X^{B}\left(t\right)=X\left(t\right)-\left(X\left(t_{i}\right)-X_{i}\right)\sigma_{ij}^{-1}\langle X\left(t\right)X\left(t_{j}\right)\rangle \;,$$
where $\sigma_{ij}=\langle X\left(t_{i}\right)X\left(t_{j}\right)\rangle$ denotes the covariance matrix. Furthermore, we imply summation over identical indices. The latter linear operation on the Gaussian random process X(t) ensures that the bridge takes on exactly the values $X_{k}$ at $t_{k}$, which can be seen from $X^{B}\left(t_{k}\right)=X\left(t_{k}\right)-\left(X\left(t_{i}\right)-X_{i}\right)\sigma_{ij}^{-1}\sigma_{kj}=X\left(t_{k}\right)-\left(X\left(t_{i}\right)-X_{i}\right)\delta_{ik}=X_{k}$, where $\delta_{ik}$ denotes the Kronecker-delta. Hence, this method allows for the reconstruction of a sparse signal where small-scale correlations are determined by the choice of the Hurst exponent *H*.

### 3.2. Phase Space Reconstruction

We first need to introduce the concept of reconstructed phase spaces [16,17], to explain the developed interpolation technique.

To choose a suitable phase-space-embedding for each data set one has to determine two parameters, the embedding dimension, and the time delay.

To estimate the time delay τ, i.e., the delay between two consecutive time steps, we employed a method based on the average information between two signals and a technique where the time delay is set using the autocorrelation-function of a signal [18].

To estimate the embedding dimension $d_{E}$, we use the algorithm of false nearest neighbors [19]. Furthermore, because the evaluations performed in this paper aim to depict the embedding space graphically, but with no limitations to the general applicability of our approach, we chose the embedding dimension to be three, i.e., $d_{E}=3$, with a corresponding value of τ=1 for all data sets. This is because, though the algorithms to determine a phase space embedding work well for deterministic chaotic systems, these algorithms sometimes yield unreasonable estimates for real-life data sets. We give all estimates for all data sets and a detailed discussion on why our choice for an embedding is reasonable in Appendix A.

The phase space embedding for a given signal $\left[x_{1},x_{2},\ldots ,x_{n}\right]$, thus is:(3)$$\vec{y}\left(i\right)\;=\;\left[x_{i},x_{i+\tau },\ldots ,x_{i+(d_{E}-1)*\tau }\right],$$
and the corresponding three-dimensional phase space embedding, thus is
(4)$$\vec{y}\left(i\right)\;=\;\left[x_{i},x_{i+\tau },x_{i+2\tau }\right].$$

### 3.3. Genetic Algorithm

We build a simple genetic algorithm to find the best possible interpolation given a time-series data’s phase space reconstruction. We want our reconstructed phase space trajectory to be as smooth as possible and thus define the trajectory’s fitness as follows.

#### 3.3.1. The Fitness of a Trajectory

The basic idea is to use a concept from image processing, i.e., measuring the blurriness of a picture, and apply it to phase space trajectories. We want our trajectory as blurry, i.e., as smooth as possible. In image processing, blurriness is determined via second-order derivatives of grey-scale images at each pixel [9]. We employ this concept, but instead of using it at each pixel, we calculate the variance of second-order derivatives along our phase space trajectories. Similar to the concept from image processing, where the low variance of second-order derivatives implies that an image is blurry, curves with a low variance of second-order derivatives exhibit comparatively smooth trajectories. The reason here is intuitively apparent. Curves with an increased variance of second-order derivatives have a range of straight and pointy sections. In contrast, curves with a low variance of second-order derivatives have a similar curvature along the trajectory and thus are smoother. Hence, in order to guarantee smoothness along the trajectory, we want this variance to be as low as possible, which thus is our loss *L*. Concluding, our fitness is maximal when our loss *L* is minimal, and the algorithm aims to achieve the lowest possible value for *L*.

Again, we start with the phase space vector and the corresponding embedding dimension $d_{E}$ and time delay τ (see Section 3.2) of each signal as
(5)$$\vec{y}\left(i\right)\;=\;\left[x_{i},x_{i+\tau },\ldots ,x_{i+(d_{E}-1)\cdot \tau }\right].$$
Thus, we have one component for each dimension of the phase space. Consequently we can write the individual components as:(6)$$y_{j}\left(i\right)=x_{i+\left(j-1\right)*\tau },$$
where $j=1,2,\ldots ,d_{E}$. We then take the second-order finite difference central derivative of a discrete function [20]:(7)$$u_{j}^{\prime \prime }\left(i\right)=x_{i+\left(j-1\right)*\tau +1}-2x_{i+\left(j-1\right)*\tau }+x_{i+\left(j-1\right)*\tau -1},\;$$
at each point, and for each component. Next, we add up all the components as:(8)$$u^{\prime \prime }\left(i\right)=\sqrt{\sum_{j=1}^{d_{E}}u_{j}^{\prime \prime }{\left(i\right)}^{2}.}$$
Furthermore, finally, we use the variance of the absolute values of second derivatives along the phase space curve as our loss *L* of a phase space trajectory:(9)$$L={\mathrm{Var}}_{i}\left[u^{\prime \prime }\left(i\right)\right].$$

#### 3.3.2. Stochastic Optimization Algorithm

The employed genetic algorithm consists of the following building blocks:

A candidate solution is an interpolated time series using a random Hurst exponent $H\in ]0;1[$. The corresponding population of candidates is, e.g., 1000 of these stochastically interpolated time series with randomly set Hurst exponents. A population of interpolated time series is generated using the multi-point Brownian bridges such that, for each member of the population, a random Hurst exponent with $H\in ]0;1[$ is chosen, which then defines the interpolation of this member of the population. After generating the population, all members are sorted with respect to their fitness, i.e., the lower the loss *L*, the better an interpolation is. The mating is implemented such that only the best 50%, with respect to fitness, can mate to produce new offspring. The mating is performed such that, for every gene, i.e., each interpolation between two data points, there is a 50:50 chance to inherit it from either one of the parents. The mutation was implemented such that, in each generation, there is a 20% chance that a randomly chosen interpolated time series is replaced with a new interpolated time series with a corresponding random Hurst exponent. Furthermore, we implemented a criterion for aborting the program, which was fulfilled if the population fitness mean did not change for ten generations. This described procedure is then performed for 1000 generations. In our numerical experiments the 1000 generations were never reached, as the criterion for abortion triggered every time, usually at approximately 200 generations.

### 3.4. The Lorenz System

For this research and to show the applicability of the proposed interpolation method, we chose the Lorenz system [21] as a model to illustrate our ideas.

The Lorenz system is a set of three nonlinear equations:(10)$$\begin{array}{cc} \frac{dx}{dt} & =10\left(-x+y\right)\;,\\ \frac{dy}{dt} & =28x-y-xz\;,\\ \frac{dz}{dt} & =xy-\frac{8}{3}z. \end{array}$$
We solved this system using a basic Runge–Kutta 4 approach [22]. We chose the step size and length of the simulation with respect to the number of interpolation points to test the quality of our interpolation scheme,
(11)$$dt=\frac{0.1}{n_{I}+1}\;,\;L=200\cdot \left(n_{I}+1\right)\;,$$
where dt is the step size and *L* is the length of the simulation. The initial conditions of the system were chosen to be:(12)x=−8,y=8,z=27.
Finally, we need a univariate signal for the phase space reconstruction, and to test our method, thus we choose one of the three variables. Accordingly, here we chose $x\left(t\right)$.

## 4. Results

Here, we present the genetic algorithm results for all data sets, first for the Lorenz system, then for five non-model data sets. For both cases, we validate the developed method such that we delete data points from the original time series and reconstruct the missing data points using the presented interpolation technique. Further, we tested the presented interpolation technique against the best random interpolation of the population, against a linear interpolation, and a cubic spline interpolation [23]. Both the linear and spline interpolation were performed using the python package scipy [24]. The employed spline interpolation from SciPy is a piecewise cubic polynomial which is twice continuously differentiable [25].

We emphasize the Lorenz system for the validation, as the generated model data allows us to test arbitrary settings, i.e., using different numbers of missing data points in accordance with the number of interpolation points. Contrary to that, for the non-model data sets, we delete every second data point and reconstruct the missing data points using the presented method. For the non-model data sets, we also present actual interpolation results, i.e., data sets with smoothed-out phase space trajectories.

### 4.1. Results for the Lorenz System

We perform our interpolation for a number of different interpolation points $N_{I}=\left\{1,2,3,4,\ldots ,20\right\}$.

We develop the following experimental steps to assess the performance of our interpolation scheme:Obtain a univariate time series from the Lorenz system.Delete points from the data which will be reconstructed later on.Given some univariate time series data of the Lorenz system, $\left[x_{1},x_{2},\ldots ,x_{n}\right]$, we extract certain data points with respect to the number of interpolation points $n_{I}\in N_{I}$ and the interpolated data set such that:
(13)$$\begin{array}{cc} \; & \left[x_{1},x_{n_{I}+2},x_{2n_{I}+2},\ldots ,x_{n}\right]\;\mathrm{Original}\;\mathrm{data}\;\mathrm{points}\;\mathrm{to}\;\mathrm{be}\;\mathrm{kept}\;\mathrm{for}\;\mathrm{interpolation},\\ \; & \left[x_{1},{\hat{x}}_{1},\ldots ,{\hat{x}}_{n_{I}},x_{n_{I}+2},{\hat{x}}_{n_{I}+1},\ldots ,{\hat{x}}_{2n_{I}+1},x_{2n_{I}+2},{\hat{x}}_{2n_{I}+2},\ldots x_{n}\right]\;\mathrm{Interpolated}\;\mathrm{data}, \end{array}$$
where ${\hat{x}}_{i}$ are the new found interpolated data points.Perform the interpolation according to the method described in Section 3.3.Calculate the RMSE for the interpolated data points with respect to the previously extracted original data points $\left[x_{2},\ldots ,x_{n_{I}+1},x_{n_{I}+3},\ldots ,x_{2n_{I}+1},\ldots \right]$. Do the same for the population mean and each time series of the initial population.
Thus, we obtain errors for the mean values of the initial population, for each time series *in* the initial population and the time series that was improved using the presented genetic algorithm. Furthermore, from all randomly generated interpolations, we select the one with the lowest RMSE to test it against the gen. alg. improved ones. Here, the root mean squared error $E_{RMSE}$, which is applied throughout this article, is given as:(14)$$E_{RMSE}\;=\;{\left(\frac{1}{n}\sum_{i=1}^{n}{\left[{\hat{x}}_{i}\;-\;x_{i}\right]}^{2}\right)}^{\frac{1}{2}}\;,$$
where $x_{i}$ are the original data points, ${\hat{x}}_{i}$ are the predicted (in this case interpolated) values and *n* is the length of the signal.

The presented results for the Lorenz system show that the algorithm can identify/generate the best interpolation in terms of a low RMSE on missing data points out of the given initial population. This can be seen in Table 1, where we highlighted the results where the genetic-algorithm-improved-interpolation outperformed every random interpolation of the population. Still, the spline interpolation outperforms the presented approach. This is also depicted in Figure 2, where we plotted the RMSE on missing data points for varying numbers of interpolation points. This graphic shows that the presented approach requires a certain amount of interpolation points, in this case, three, to be close to the best random interpolation of the population. We assume that the reason for this is that the variance of second derivatives along a phase space trajectory requires a certain *density* of phase space points to be able to differ between smooth and edgy phase space trajectories. On the other hand, the spline interpolation performs well right from the start.

The corresponding reconstructed phase space plots (Figure 3) show that both the best random interpolation (e) and the genetic-algorithm-improved interpolation (f) provide convincing phase space portraits, as both are indeed close to the ground truth (a). On the other hand, the population mean (b) is far off and features many sharp edges and pointy sections. Furthermore, the linear interpolation (c) provides a very edgy phase-space portrait, just as one would expect from linear interpolation. Contrary to that, from all presented phase space portraits, the one for the spline interpolation (d) is most similar to the original one, i.e., even the initial abbreviations caused by the time delay are almost perfectly reconstructed.

We further plotted all obtained results for 13 interpolation points as time series in Figure 4. The results show that the population mean (a) is far off the ground truth and differs drastically at the high and low peaks, as it does not reach the actual data points. Both the genetic-algorithm-improved (b) and the best random interpolation of the initial population (c) capture most of the high and low peaks compared to the population mean. Further, when comparing the genetic-algorithm-improved and the population mean (d), one can see that the improved interpolation provides a smoother curve when depicted as a time series. In contrast, the population mean tends to produce sharp peaks. Finally, we compare the linear interpolation (e), and the spline interpolation (f) to the genetic algorithm improved interpolation. The linear interpolation here is far off, but the spline interpolation reproduces the Lorenz system almost perfectly and thus outperforms the genetic-algorithm-improved interpolation.

### 4.2. Results for Non-Model Data Sets

This section tests our interpolation scheme on real-life data sets with only a limited number of sampled data points. However, these are the focus of the proposed method, i.e., to increase the fine-grainedness of short, sparsely-sampled time series data, e.g., environmental or agricultural data sets. We must stress that our method is not restricted to equidistant time series: Due to the general form of the bridge construction (Equation 2), non-equidistant time series excerpts can be interpolated as well.

For this reason, we chose five data sets to demonstrate our method further, i.e., we validate the interpolation with missing data points and then present an actual interpolation and the improved phase space trajectories for each time-series data. We consider a phase space trajectory to be improved if we achieve smoother trajectories, which exhibit fewer edgy points in a phase space representation. Further, we rescaled every data set to the unit interval and subtracted a linear fit from the data set if a linear trend was clearly visible for the reconstructed phase space plots.

The validation on these non-model data sets is performed such that every second data point of the original time series is deleted. Then, all the gaps are interpolated to reconstruct the missing data points. The results are shown for the average prediction of the population, the random interpolation with the lowest RMSE, a linear interpolation, a cubic spline interpolation, and the improved interpolation using the presented genetic algorithm. This section features only the validation errors; the corresponding plots are collected in Appendix B to keep the main text focused.

#### 4.2.1. NYC Measles Outbreaks

This is a data set that we obtained from [26], where it is discussed and shown to feature an attractor structure in the embedded phase space. The corresponding original source is [27]. It depicts measles outbreaks in New York City (NYC) from 1928 to 1964, binned every two weeks, with a total of 432 data points. The data set depicts sharp repetitive peaks, i.e., the increase and decrease of measles cases in NYC. Due to its regularity, we expect PhaSpaSto-interpolation to perform well.

The results on how well the presented interpolation can reproduce missing data points of this data set are collected in Table 2 and depicted in Figure A8a. These results show that, though the genetic-algorithm-improved interpolation drastically outperforms the average random interpolation, the algorithm did not once outperform the best interpolation of the population. Still, starting with seven interpolation points, the genetic-algorithm-improved interpolation performs well and is very close to the best of 1000 randomly interpolated results, i.e., always below or around the best 1% of the population. Further, *PhaSpaSto*-interpolation does outperform the cubic spline interpolation starting with five interpolation points. We thus conclude that the presented interpolation technique captures the phase-space properties of this data set and effectively can be used to interpolate this time series. Furthermore, compared to the cubic and linear interpolation, the proposed method requires at least seven interpolation points to reach peak performance for this data set. All validation plots, are collected in Section B.2.

An interpolation of the original data set is depicted in Figure 5. Comparing the reconstructed phase space of the original data set, the population mean (c), and the presented interpolation technique (d); we see that the phase space portrait of the latter features a smoothed-out phase space trajectory compared to the original time series (b) and the population mean (c), which are both pointy and contain many sharp edges. Further, considering the graph of the actual time series (a), we see that the presented interpolation technique increases the major peaks, thus making extreme events more prominent.

#### 4.2.2. Car Sales in Quebec

This is a data set from the Time Series Data Library [28]. It depicts monthly car sales in Quebec from January 1960 to December 1968, with a total of 108 data points. This data set clearly shows an increasing linear trend and oscillatory regularities, i.e., seasonal behavior.

The results on the reproducibility of missing data points for all interpolation techniques are collected in Table 3, and depicted in Figure A8b. The genetic-algorithm-improved interpolation drastically outperforms the average random interpolation. Further, the *PhaSpaSto*-interpolation always outperforms the cubic spline and linear interpolation. The overall best performance is mostly achieved by a random interpolation, still the PhaSpaSto-interpolation performs best for one, three, and five interpolation points. Overall, the genetic-algorithm-improved interpolation performs well and is very close to the best of 1000 randomly interpolated results, i.e., for most cases below or around the best 1% of the population. Thus, we conclude that the presented interpolation technique effectively captures the phase-space properties of this data set and can be used to interpolate this time series data. All additional plots for the validation are collected in Section B.3, where one can find the reconstructed attractors for all interpolated validation sets and the corresponding time series plots.

An interpolation of the original data set is depicted in Figure 6. Here Figure 6c,d present the population mean and the improved interpolation, respectively. When comparing them, one can see that the genetic algorithm improves the phase space portrait in terms of a smoothed-out phase space trajectory compared to the original time series (b) and the population mean (c), which are both pointy and have many sharp edges. When considering the actual time-series graph (a), the presented interpolation technique increases the major peaks, thus making extreme events more prominent. Further, it provides a rather smooth curve, i.e., no pointy edges, as depicted in the zoomed-in plot in (a).

#### 4.2.3. Perrin Freres Champagne Sales

This is a data set from the Time Series Data Library [28]. It depicts Perrin Freres Champagne sales from January 1964 to September 1972, with a total of 105 data points.

The validation results for this data set are collected in Table 4 and Figure A8c.

Though the genetic-algorithm-improved interpolation drastically outperforms the average random interpolation, the algorithm did not once outperform the best interpolation of the population. Still, starting with five interpolation points, the genetic-algorithm-improved interpolation performs well and is very close to the best of 1000 randomly interpolated results, i.e., consistently below or around the best 1% of the population. Overall the cubic spline interpolation performed best on this data set. The linear interpolation, though outperforming the population mean, is still far off. We thus conclude that the presented interpolation technique does capture the phase-space properties of this data set from a given population and can be used to interpolate this time series data, but the cubic spline interpolation is the better choice.

An interpolation of the original data set is depicted in Figure 7. We again show the population mean (c) and the improved interpolation (d). The presented interpolation technique improves the phase space portrait in terms of a smoothed-out phase space trajectory (d) compared to the original time series (b) and the population mean (c), which are both pointy and contain many sharp edges. Here the population mean increased sharp edges drastically. Further, considering the graph of the actual time series (a), the presented interpolation technique increases the major peaks, thus making extreme events more prominent and providing a thoroughly smooth curve, as depicted in the zoom-in window in (a).

#### 4.2.4. Monthly Airline Passengers

This is a data set from the Time Series Data Library [28]. It depicts monthly international airline passengers from January 1949 to December 1960, with 144 data points, given in units of 1000. Again, this data set shows a visible linear trend, and strong seasonal oscillatory regularities.

The results on how well the presented interpolation can reproduce missing data points of this data set are collected in Table 5 and depicted in Figure A8d. The results show that, though the genetic-algorithm-improved interpolation drastically outperforms the average random interpolation, the algorithm did not once outperform the best interpolation of the population. Still, starting with three interpolation points, the algorithm did outperform both the linear and the cubic spline interpolation. What is curious, though, is that, for this data set, of all the non-model data sets, the linear interpolation outperforms the cubic spline interpolation.

The genetic-algorithm-improved interpolation does not perform that well for this data set compared to a random interpolation of the time series. The improved interpolation is only around the best ≈40% of the initial population for this data set. We thus conclude that the presented interpolation technique does not capture the phase-space properties of this data set very well. Still, the genetic algorithm does improve the initial population such that the population mean, the linear interpolation, and the cubic spline interpolation are outperformed, starting with three interpolation points. All-time series and reconstructed attractor plots for this data set can be found in Section B.5.

An actual interpolation of the original data set is depicted in Figure 8. We again show the population mean (c) and the improved interpolation (d). The presented *PhaSpaSto*-interpolation (d) improves the phase space portrait in terms of a smoothed-out phase space trajectory, compared to the original time series (b) and the population mean (c), which are both pointy and contain many sharp edges. Further, considering the actual time series (a) graph, PhaSpaSto-interpolation technique slightly increases the major peaks. Furthermore, compared to the other non-model data sets, the improved interpolation does provide a relatively smooth curve, but it appears much sharper than for, e.g., the car sales in Quebec data set (see Figure 6a).

#### 4.2.5. Monthly Mean Temperature in Nottingham Castle

This is a data set from the Time Series Data Library, [28]. It depicts the mean monthly temperature in Nottingham castle from January 1920 to December 1939, given in degrees Fahrenheit, with a total of 240 data points. This data set shows strong seasonal regularities and behaves stationary, as no linearly increasing or decreasing trend is visible.

The results on how well the presented interpolation can reproduce missing data points of this data set are collected in Table 6 and depicted in Figure A8e. The results show that, though the genetic-algorithm-improved interpolation drastically outperforms the average random interpolation, the algorithm did not once outperform the best interpolation of the population, although outperforming the linear and the cubic spline interpolation. The genetic-algorithm-improved interpolation does not perform that well for this data set compared to a random interpolation of the time series. The improved interpolation is only around the best ≈34% for this data set. We thus conclude that the presented interpolation technique does not capture the phase-space properties of this data set very well. The corresponding time-series and reconstructed phase space plots are collected in Section B.6.

An interpolation of the original data set is depicted in Figure 9. We again show the population mean (c) and the improved interpolation (d). The presented interpolation technique improves the phase space portrait (d) in terms of a smoothed-out phase space trajectory compared to the original time series (b) and the population mean (c), which are both pointy and contain many sharp edges. Furthermore, given the time-series depiction of the *PhaSpaSto*-interpolation (Figure 9a), we see the same behavior as for all the other data sets; the major peaks are increased.

#### 4.2.6. Monthly Shampoo Sales

This is a data set from the Time Series Data Library [28]. This data set describes monthly shampoo sales over three years, i.e., 36 observations. Although this data set clearly shows a linear trend, no obvious regularities or seasonalities are apparent. Thus, we consider this data set to be a more stochastical rather than an oscillatory one. We expect PhaSpaSto-interpolation to not perform well on data sets such as these.

Table 7 and Figure A8f both show the results on how well the employed interpolation techniques can reconstruct missing data points on this data set. For this data set, PhaSpaSto-interpolation does not perform well at all. The best performance is achieved by the random interpolation with the lowest error, followed by the population mean and the linear interpolation. Spline interpolation performs worst on this data set. Because of its stochastic nature and no apparent seasonalities, PhaSpaSto-interpolation is not a well-suited method for interpolating this data set. The corresponding time-series and reconstructed phase space plots are collected in Section B.7.

entropy-24-00718-t007_Table 7Table 7Errors for the interpolated data on the shampoo sales data set depending on the number of interpolation points. The errors are shown for the mean interpolation of all populations, the lowest error in the population, and the interpolation improved using the presented genetic algorithm. We highlighted the interpolation where the genetic-algorithm-based interpolation performed best. The corresponding plots for the best interpolation are shown in Section B.7. Further, we give the percentage of the population that is outperformed by the genetic-algorithm-improved interpolation.

nI


**1**

**3**

**5**

**7**

**9**

**11**

**13**

**15**

**RMSE population mean**
93.7798593.7809693.7779493.7847493.7885293.7807893.7828593.78171
**Lowest RMSE in population**
75.1755375.0149375.0448875.1351175.0285475.2032975.0844575.12778
**RMSE linear interpolated**
100.56301100.56301100.56301100.56301100.56301100.56301100.56301100.56301
**RMSE spline inteprolated**
108.02059108.02059108.02059108.02059108.02059108.02059108.02059108.02059
**RMSE gen. alg. improved**
110.19816105.25685106.31509105.51261105.50765103.98034105.90568105.86858
**Below Best %**
99.5%99.4%99.5%99.5%99.5%99.599.5%99.5%

nI


**17**

**19**

**21**

**23**

**25**

**27**

**29**

**31**

**RMSE population mean**
93.7858793.7810893.77988
**93.78166**
93.7844493.7797293.7821093.78348
**Lowest RMSE in population**
75.1290375.0810175.26534
**75.15750**
75.1270675.1762675.0362575.05900
**RMSE linear interpolated**
100.56301100.56301100.56301
**100.56301**
100.56301100.56301100.56301100.56301
**RMSE spline inteprolated**
108.02059108.02059108.02059
**108.02059**
108.02059108.02059108.02059108.02059
**RMSE gen. alg. improved**
105.44456105.51313105.51292
**103.69012**
104.71252104.01555103.73095103.88873
**Below Best %**
99.5%99.5%99.5%
**99.5%**
99.5%99.599.5%99.5%

An interpolation of the original data set is depicted in Figure 10. We again show the population mean (c) and the improved interpolation (d). The presented interpolation technique improves the phase space portrait (d) in terms of a smoothed-out phase space trajectory compared to the original time series (b) and the population mean (c), which are both pointy and contain many sharp edges. Furthermore, given the time-series depiction of the *PhaSpaSto*-interpolation (Figure 10a), we see that PhaSpaSto-interpolation slightly increases some of the major peaks, but overall presents an interpolation similar to what we would expect from a spline interpolation of the data set.

#### 4.2.7. Annual Maize Yields in Austria

This is a data set of the annual yields of maize in Austria ranging from 1961 to 2017 with an overall of 57 data points. This data set can be downloaded at http://www.fao.org/faostat/, accessed on 15 May 2022. as with the shampoo sales data set, this data set does not provide us with visible seasonalities but an overall random behavior and a visible increasing linear trend.

The results on how well the presented interpolation can reproduce missing data points of this data set are collected in Table 8 and depicted in Figure A8g. PhaSpaSto-interpolation performs second-worst on this data set. Spline interpolation performs worst, and the random interpolation with the lowest error performs best. The second-best is the average interpolation of all random interpolations. The third-best is the linear interpolation, thus concluding that a random or a linear interpolation is a better choice on data sets with no apparent trends. The corresponding time-series and reconstructed phase space plots are collected in Section B.8.

An interpolation of the original data set is depicted in Figure 11. We again show the population mean (c) and the improved interpolation (d). The presented interpolation technique improves the phase space portrait (d) in terms of a smoothed-out phase space trajectory compared to the original time series (b) and the population mean (c), which are both pointy and contain many sharp edges. Furthermore, given the time-series depiction of the *PhaSpaSto*-interpolation (Figure 11a), we see similar behavior as for all the other data sets, some major peaks are increased, and overall the interpolation is very much how one would expect a spline interpolation to appear.

### 4.3. Summary

We briefly summarize this research and highlight the main findings:We presented a genetic algorithm to improve a stochastic interpolation, i.e., multi-point fractional Brownian bridges, to enhance the reconstructed phase space of any given time series. For simplicity, we named this method *PhaSpaSto*-interpolation.We presented a novel approach to measure the quality of a phase space reconstruction according to Takens’ theorem. Here we used an idea from image processing, i.e., to identify blurry images via the variance of second derivatives. These second derivatives are calculated along the reconstructed phase space curve for any given reconstructed phase space. We use the variance of these second derivatives to measure the quality of our phase space reconstruction. Given two interpolated phase space curves of the same time series, the one with the lower variance of second derivatives along the curve is the better phase space reconstruction, as it is the smoother one.We showed that the developed technique performed well in the case of a model data set, i.e., one variable of the Lorenz system. Here, we deleted data points from the original time series data and were able to outperform, in some cases, any best random interpolations of this time series data. Furthermore, the presented method outperformed a linear interpolation when locating the missing data points. Still, the proposed method did not outperform the presented cubic spline interpolation on this task. This was conducted to validate our method and to show its applicability. Further, the presented reconstructed phase spaces plots show that the interpolated phase space reconstruction is similar to the original reconstructed phase space. The results for the Lorenz system are collected in Section 4.1.We demonstrated the presented method using seven sparsely sampled non-model data sets. The validation was performed such that we deleted every second data point from the original time series and reconstructed these missing data points using the developed technique. For three out of seven data sets, the developed method effectively can identify the interpolations or parts of it with low errors, i.e., the result is around the best 1% of the population in terms of the RMSE for the reconstructed data points. *PhaSpaSto*-interpolation outperformed the spline interpolation for six of seven non-model data sets and the linear interpolation on five non-model data sets. Furthermore, the best random interpolation outperformed the cubic spline interpolation on six non-model data sets. For the monthly airline passengers data set, the *PhaSpaSto*-interpolation does not perform very well as it is only around the best 30–40% of all RMSEs of the population. Furthermore, for the final two data sets, i.e., the shampoo sales and maize yields data sets, PhaSpaSto-interpolation cannot find a meaningful interpolation and is outperformed by every other interpolation except the cubic spline interpolation. The interpolation performed well in case of the measles cases in NYC data set (Section 4.2.1), the car sales in Quebec data set (Section 4.2.2) and the Perrin Freres champagne sales data set (Section 4.2.3), which are data sets that show regularities and an oscillatory behavior. The cases where the presented method did not perform well are the monthly international airline passengers data set (Section 4.2.4), the monthly mean temperature in Nottingham castle data set (Section 4.2.5), the shampoo sales data set and the annual maize yields in Austria data set. We conclude that PhaSpaSto-interpolation can retrieve missing data points for time series with seasonal behavior or oscillatory regularities better than for stochastic data sets.We also used the seven non-model data sets to show the applicability of the developed technique as an actual interpolation technique, i.e., no deleted data points. The plots of the reconstructed phase spaces show that it softens the edges and provides a thoroughly smoother and cleaner reconstructed phase space trajectory. Therefore, the authors conclude that this technique applies to arbitrary univariate data sets. All of these plots are collected in Section 4.2. We further recommend it when dealing with sparsely-sampled seasonal time series, or time series that show oscillatory regularities.

## 5. Conclusions

This article presents a novel approach to interpolate univariate time series data. For simplicity, we named this method *PhaSpaSto*-interpolation. The concept is first to generate a population of, e.g., 1000, different stochastically-interpolated time series data. This is performed using multi-point Brownian bridges, each assigned with a random Hurst exponent. Then, as a second step, a genetic algorithm generates one time series out of the population with a low variance of second-order derivatives along the corresponding reconstructed phase space trajectory, i.e., we want this curve to be as smooth as possible. The idea of using the variance of second-order derivatives is adapted from image processing, where the variance of second-order derivatives is used to differentiate between blurry and sharp images. We also tested the discussed approach with different loss functions that, in the end, did not work. These failed attempts are collected in Appendix C. Thus, we want to point out the variance of second-order derivatives as a valuable tool for phase space analysis and should be considered when dealing with non-linear time series data.

We then applied the presented interpolation technique to the Lorenz system, or to be specific, to one of the variables of the Lorenz system, since we are dealing with univariate data only. Next, we deleted data points from this time series and interpolated the missing data points with the presented interpolation technique, i.e., interpolated subsampled time series. We also tested the proposed approach against linear interpolation and cubic spline interpolation. The results show that the presented PhaSpaSto-interpolation can reproduce the Lorenz system, i.e., the genetic algorithm can find the best parts of the initial population to reconstruct the Lorenz system. Still, the spline interpolation outperformed the *PhaSpaSto*-interpolation for the Lorenz system.

Finally, we applied the presented approach to various real-life and/or benchmark data sets. There are no fine-grained model data available for data sets such as these. We cannot verify the interpolation as we did with the Lorenz system. Instead, we deleted every second data point of these data sets. We reconstructed them using the developed method, i.e., generated interpolations using a range of interpolation points, selecting the missing data points, and verifying them against the ground truth. *PhaSpaSto*-interpolation performed well on three of seven data sets, as the genetic algorithm can identify/build interpolations with low errors for the missing data points.

Further, *PhaSpaSto*-interpolation outperformed the spline-interpolation on six of seven data sets. Thus, we conclude that the presented method can also be applied to non-model data sets but performs best on seasonal data or data with oscillatory regularities. For inherently random data sets, we recommend using multi-point Brownian bridges instead. Lastly, we show actual interpolations on these non-model data sets, i.e., no deleted data points. Given that the reader is familiar with how strange attractors of chaotic systems appear, it should be clear from the presented reconstructed phase space portraits that our approach can interpolate real-life data as one would expect a phase space embedding of a strange attractor to appear, see Section 4.2. Future research will also be devoted to generalizations of the bridge process (Equation 2) to random processes which exhibit non-Gaussian features [29].

We expect the presented research to be useful for predicting and analyzing sparsely-sampled time series data, e.g., in agriculture or other fields where fine-grained measurements are expensive. Furthermore, our methodology could be applicable to a broad range of other real-world problems such as the filling of gaps in solar wind measurements [2] or spatiotemporal wind fields [3] for the assessment of wind turbine loads. As our method can be considered as some “hybrid” between a stochastic (by virtue of the fractional Brownian bridge interpolation) and a deterministic algorithm (by the embedding and genetic algorithm), it should be highly relevant for the filling of such time series or spatial fields, which often exhibit deterministic and stochastic elements at the same time.

We further expect the presented research to be utilized for improving machine and deep learning approaches with insufficient data. Thereby, an enhanced phase space structure might improve a forecast’s accuracy. Thus, the variance of second derivatives along reconstructed phase space trajectories can be used as an indicator for the quality of a prediction.

Future improvements and applications of this technique include the expansion to multi-variate data sets and using the presented loss function, i.e., the variance of second derivatives along phase space trajectories, to find better phase space embeddings. The interested reader is therefore referred to Appendix D where we present the loss-surface of the Lorenz system with a varying time delay τ and a varying embedding dimension $d_{E}$.

Furthermore, as previously mentioned, we wish to test to what degree improved phase-space embeddings can be beneficial for machine and/or deep learning approaches for learning and predicting time-series data.

Lastly, the presented interpolation technique code will be available on GitHub from the corresponding author in the future.

## Figures and Tables

**Figure 1 entropy-24-00718-f001:**
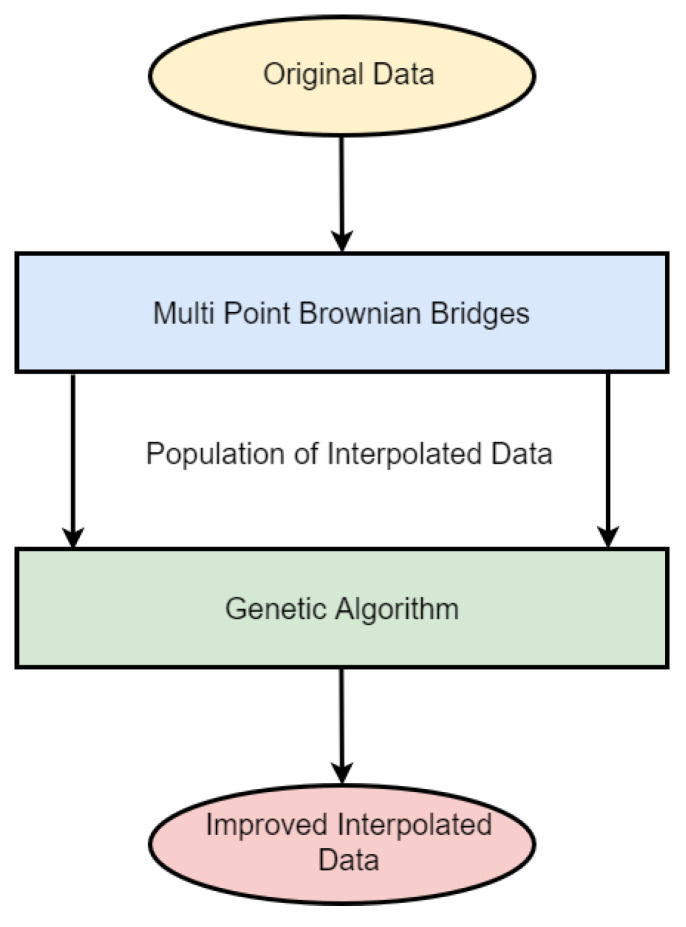
Depiction of the employed scheme.

**Figure 2 entropy-24-00718-f002:**
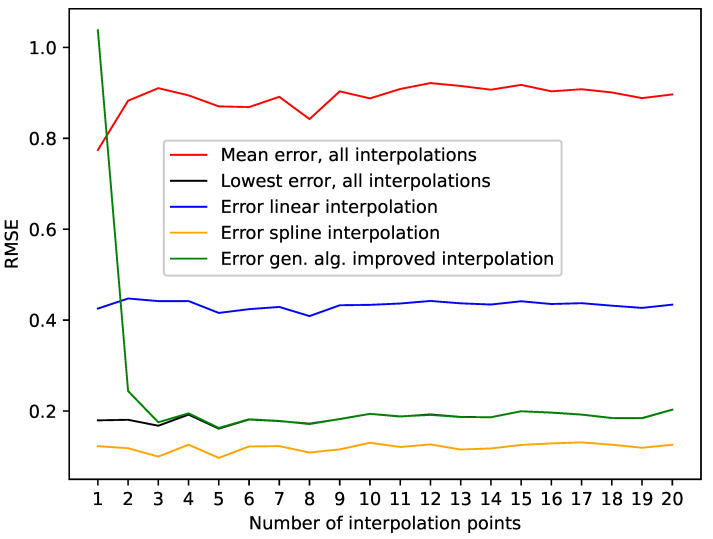
Errors from Table 1 depending on the different numbers of interpolation points.

**Figure 3 entropy-24-00718-f003:**
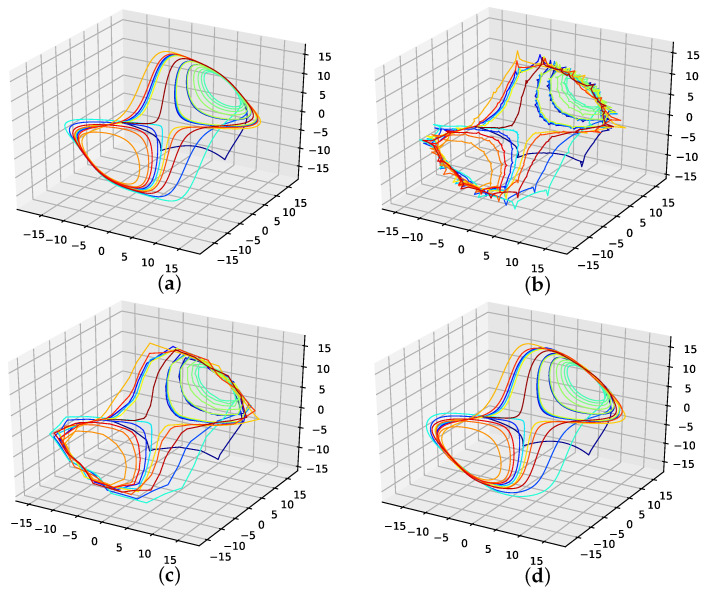
Reconstructed attractors for the interpolated Lorenz system. (**a**): Non-interpolated original data (i.e., the one the errors are calculated with); (**b**): Average interpolation of the whole population; (**c**): Linear interpolated; (**d**): Spline interpolated; (**e**): The one interpolation of the population that has the lowest RMSE; (**f**): Interpolation improved by the presented genetic algorithm approach.

**Figure 4 entropy-24-00718-f004:**
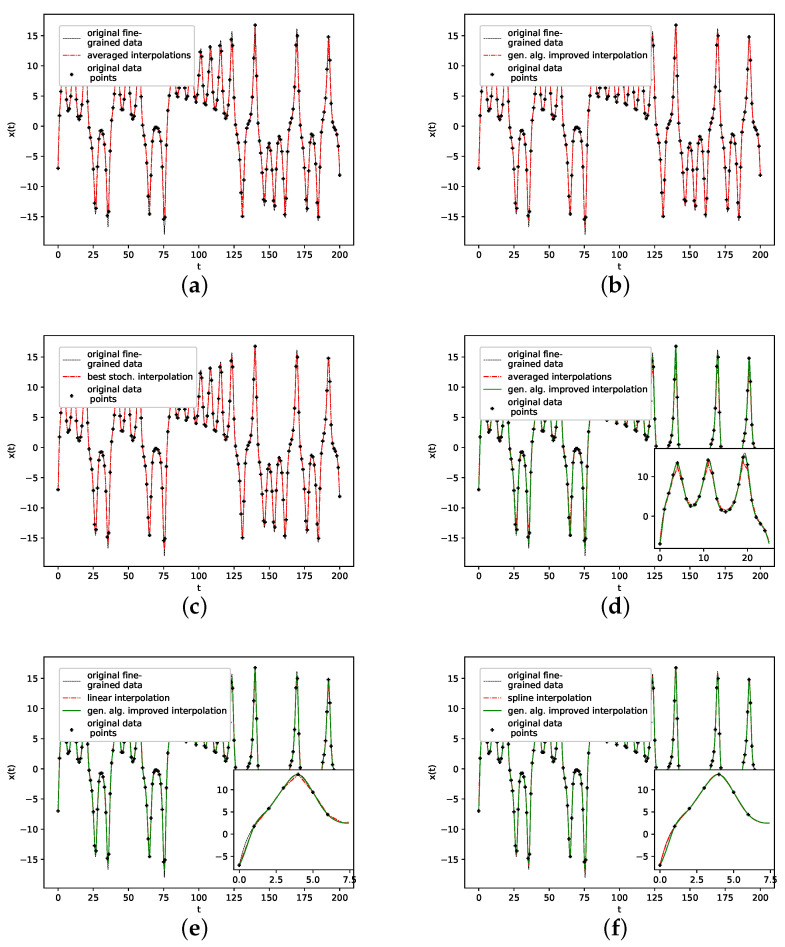
Original vs. interpolated time series data. (**a**): Non-interpolated original data (i.e., the one the error’s are calculated with) and population average; (**b**): Genetic-algorithm-improved interpolation; (**c**): The one interpolation of the population that has the lowest RMSE; (**d**): Population average vs. genetic-algorithm-improved interpolation; (**e**): Linear interpoaltion vs. genetic-algorithm-improved interpolation; (**f**): Spline interpolation vs. genetic-algorithm-improved interpolation.

**Figure 5 entropy-24-00718-f005:**
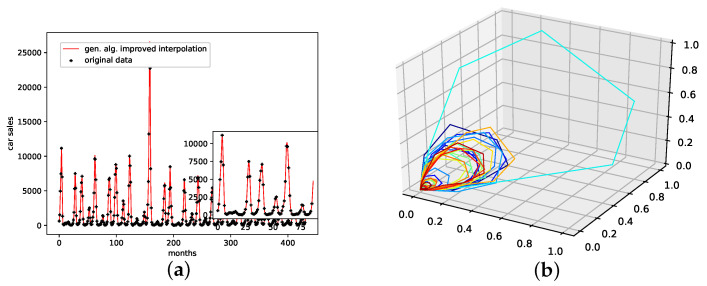
Interpolated data and reconstructed attractors for the NYC measles outbreaks data set. (**a**): The original and interpolated time series data; (**b**): Phase space reconstruction of the original data; (**c**): Phase space reconstruction of the average population data; (**d**): Phase space reconstruction of the genetic-algorithm-improved data.

**Figure 6 entropy-24-00718-f006:**
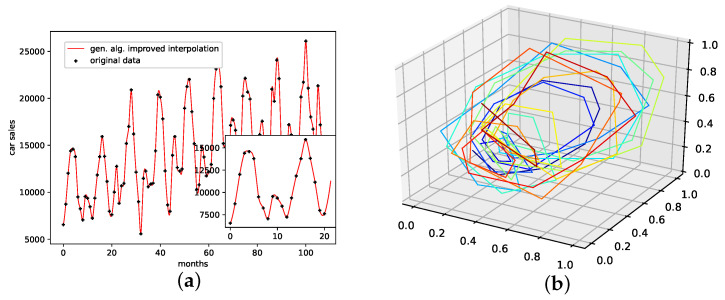
Interpolated data and reconstructed attractors for the car sales in Quebec data set. (**a**): The original and interpolated time series data; (**b**): Phase space reconstruction of the original data; (**c**): Phase space reconstruction of the average population data; (**d**): Phase space reconstruction of the genetic-algorithm-improved data.

**Figure 7 entropy-24-00718-f007:**
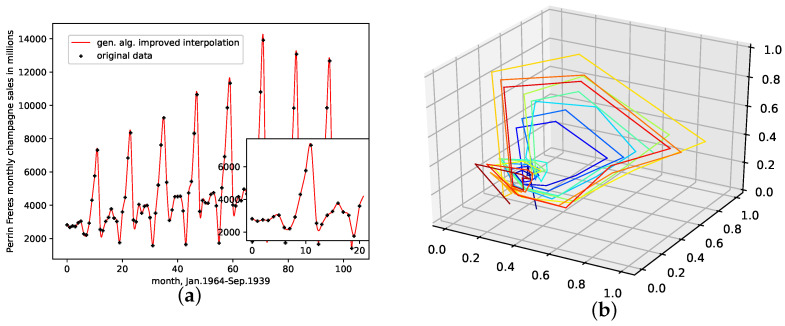
Interpolated data and reconstructed attractors for the Perrin Freres Champagne sales data set. (**a**): The original and interpolated time series data; (**b**): Phase space reconstruction of the original data; (**c**): Phase space reconstruction of the average population data; (**d**): Phase space reconstruction of the genetic-algorithm-improved data.

**Figure 8 entropy-24-00718-f008:**
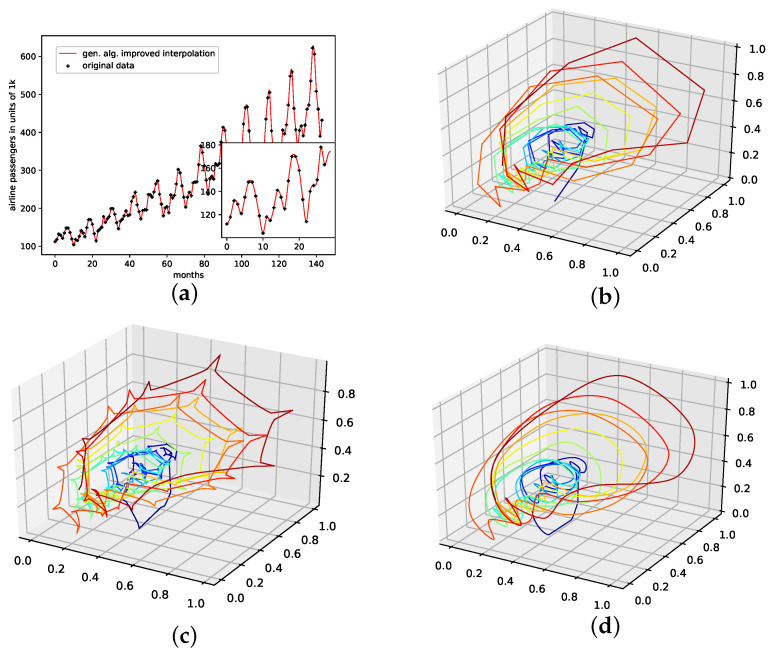
Interpolated data and reconstructed attractors for the monthly international airline passengers data set. (**a**): The original and interpolated time series data; (**b**): Phase space reconstruction of the original data; (**c**): Phase space reconstruction of the average population data; (**d**): Phase space reconstruction of the genetic-algorithm-improved data.

**Figure 9 entropy-24-00718-f009:**
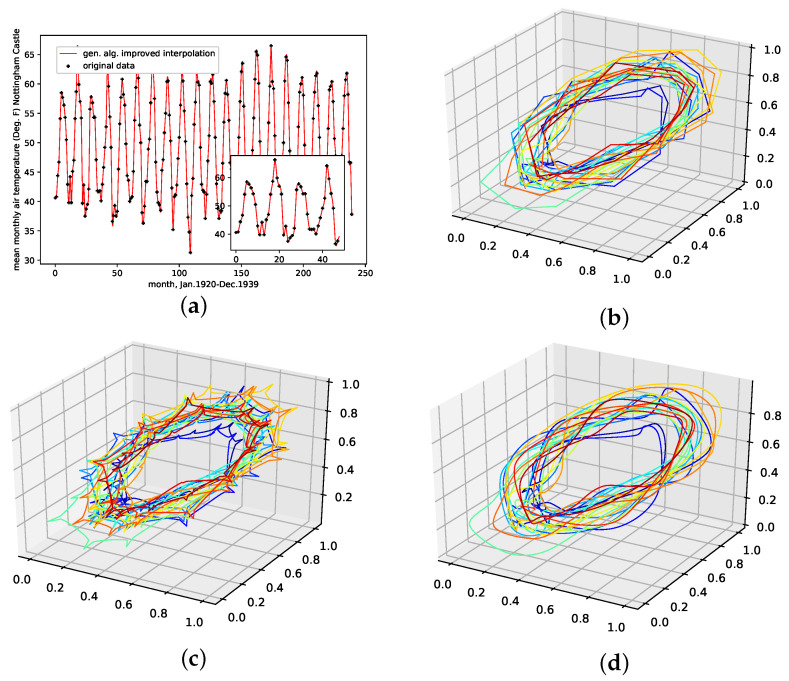
Interpolated data and reconstructed attractors for the monthly mean temperature in Nottingham castle data set. (**a**): The original and interpolated time series data; (**b**): Phase space reconstruction of the original data; (**c**): Phase space reconstruction of the average population data; (**d**): Phase space reconstruction of the genetic-algorithm-improved data.

**Figure 10 entropy-24-00718-f010:**
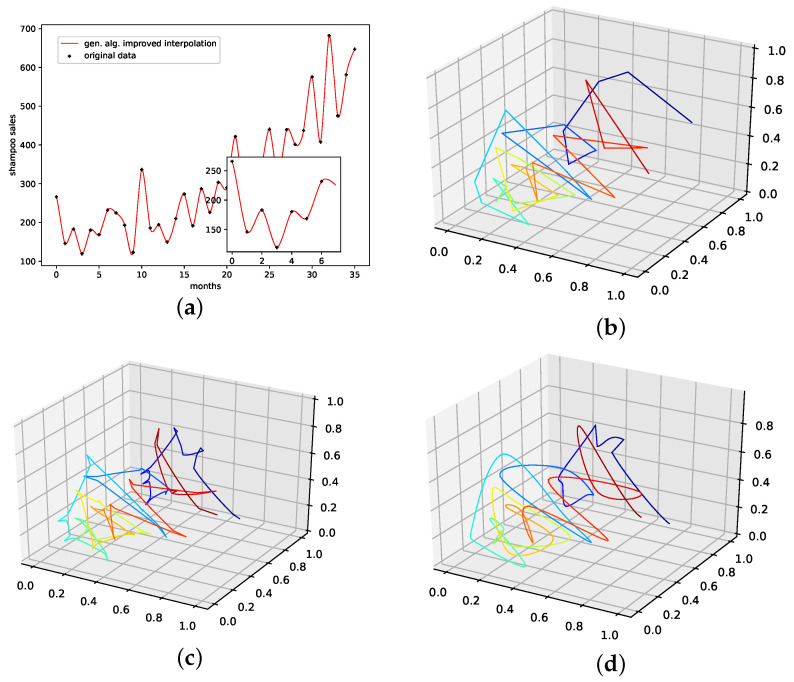
Interpolated data and reconstructed attractors for the shampoo sales data set. (**a**): The original and interpolated time series data; (**b**): Phase space reconstruction of the original data; (**c**): Phase space reconstruction of the average population data; (**d**): Phase space reconstruction of the genetic-algorithm-improved data.

**Figure 11 entropy-24-00718-f011:**
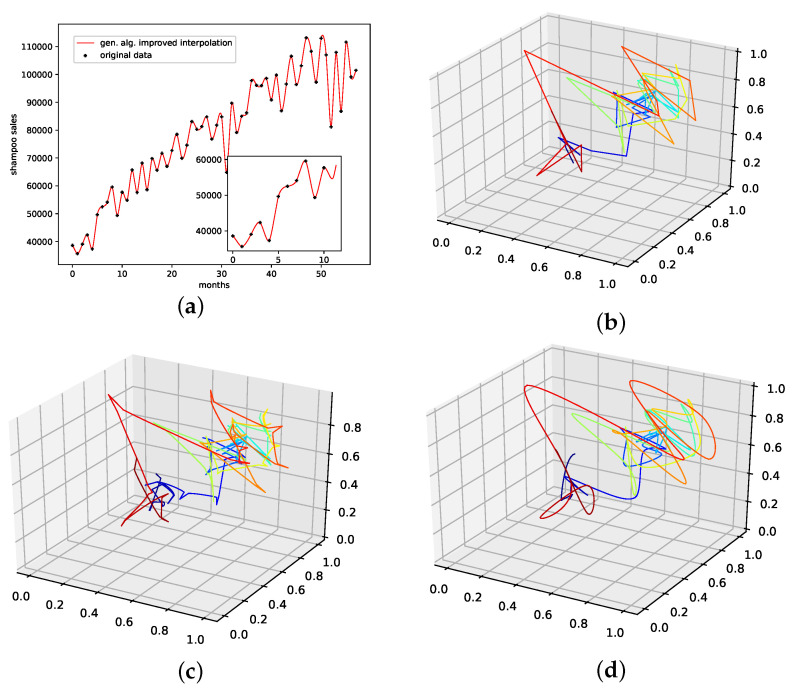
Interpolated data and reconstructed attractors for the annual maize yields in Austria data set. (**a**): The original and interpolated time series data; (**b**): Phase space reconstruction of the original data; (**c**): Phase space reconstruction of the average population data; (**d**): Phase space reconstruction of the genetic-algorithm-improved data.

**Table 1 entropy-24-00718-t001:** Errors for the interpolated data on the Lorenz system depending on the number of interpolation points. The errors are shown for the mean interpolation of all populations and improved interpolation using the presented genetic algorithm. *Lowest RMSE in population* refers to the best randomly interpolated result, i.e., the one interpolation from the population that produced the lowest error by chance. We also featured the results for the linear and spline interpolation. We highlighted the interpolations where the genetic-algorithm-based interpolation outperformed the whole population of interpolations. Further, we give the percentage of the population outperformed by the genetic-algorithm-improved interpolation. This table is depicted in Figure 2.

$n_{I}$	1	2	3	4	5	6	7	8	9	10
RMSE population mean	0.77419	0.88263	0.91026	0.89442	0.87013	0.86858	0.89120	**0.84220**	0.90323	0.88777
lowest RMSE in population	0.17939	0.18068	0.16757	0.19206	0.16126	0.18134	0.17782	**0.17211**	0.18216	0.19371
RMSE linear interpolated	0.42534	0.44752	0.44179	0.44185	0.41574	0.42406	0.42894	**0.40883**	0.43263	0.43353
RMSE spline interpolated	0.12263	0.11808	0.09968	0.12586	0.09678	0.12195	0.12280	**0.10862**	0.11554	0.13008
RMSE gen. alg. improved	1.03779	0.24381	0.17517	0.19488	0.16264	0.18182	0.17818	**0.17121**	0.18239	0.19374
below best %	74.4%	21.6%	4.3%	2.2%	1.4%	1.1%	0.7%	**0.1%**	0.8%	0.3%
$n_{I}$	**11**	**12**	**13**	**14**	**15**	**16**	**17**	**18**	**19**	**20**
RMSE population mean	0.90844	**0.92145**	**0.91509**	**0.90686**	**0.91750**	0.90326	0.90789	0.90080	0.88835	0.89651
lowest RMSE in population	0.18789	**0.19238**	**0.18693**	**0.18632**	**0.19943**	0.19640	0.19208	0.18449	0.18415	0.20291
RMSE linear interpolated	0.43649	**0.44211**	**0.43687**	**0.43423**	**0.44142**	0.43534	0.43720	0.43170	0.42685	0.43398
RMSE spline interpolated	0.12086	**0.12646**	**0.11530**	**0.11765**	**0.12532**	0.12873	0.13098	0.12581	0.11912	0.12581
RMSE gen. alg. improved	0.18816	**0.19141**	**0.18670**	**0.18626**	**0.19943**	0.19663	0.19215	0.18462	0.18441	0.20300
below best %	0.8%	**0.1%**	**0.1%**	**0.1%**	**0.1%**	0.8%	0.6%	0.6%	0.8%	0.6%

**Table 2 entropy-24-00718-t002:** Errors for the interpolated data on the NYC measles data set depending on the number of interpolation points. The errors are shown for the mean interpolation of all populations, the lowest error in the population, and the interpolation improved using the presented genetic algorithm. We highlighted the interpolation where the genetic-algorithm-based interpolation performed best. The corresponding plots for the best interpolation are shown in Section B.2. Further, we give the percentage of the population outperformed by the genetic-algorithm-improved interpolation.

$n_{I}$	1	3	5	7	9	11	13	15
**RMSE** **Population** **Mean**	860.56140	860.56165	860.56098	860.56235	860.56124	860.56210	860.56145	860.56220
**Lowest** **RMSE in** **population**	594.27833	594.27832	594.27832	594.27832	594.27748	594.27831	594.27832	594.27833
**RMSE linear** **interpolated**	713.61079	713.61089	713.61089	713.61089	713.61089	713.61089	713.61089	713.61089
** $n_{I}$ **	**1**	**3**	**5**	**7**	**9**	**11**	**13**	**15**
**RMSE spline** **interpolated**	607.03778	607.03778	607.03778	607.03778	607.03778	607.03778	607.03778	607.03778
**RMSE** **gen. alg.** **improved**	1138.28460	621.70136	602.03361	594.36367	594.33054	594.34891	594.34819	594.34132
**Below** **Best %**	75.3%	25.8%	13.4%	0.8%	0.8%	0.8%	0.8%	0.8%
** $n_{I}$ **	**17**	**19**	**21**	**23**	**25**	**27**	**29**	**31**
**RMSE** **Population** **Mean**	860.56196	860.56132	860.56168	860.56090	**860.56153**	860.56287	860.56138	860.56192
**Lowest** **RMSE in** **population**	594.27901	594.27750	594.27934	594.27834	**594.28039**	594.27831	594.28069	594.27833
**RMSE linear** **interpolated**	713.61089	713.61089	713.61089	713.61089	**713.61089**	713.61089	713.61089	713.61089
**RMSE spline** **interpolated**	607.03778	607.03778	607.03778	607.03778	**607.03778**	607.03778	607.03778	607.03778
**RMSE** **gen. alg.** **improved**	594.33772	594.33837	594.33400	594.35508	**594.31806**	594.36050	594.42183	594.39145
**Below** **Best %**	0.8%	0.8%	0.8%	0.8%	**0.6%**	0.8%	1.1%	1.1%

**Table 3 entropy-24-00718-t003:** Errors for the interpolated data on the car sales in Quebec data set depending on the number of interpolation points. The errors are shown for the mean interpolation of all populations, the linear interpolation, the cubic spline interpolation, as well as for the lowest error in the population, and for the interpolation that was improved using the presented genetic algorithm. We highlighted the interpolation where the genetic-algorithm-based interpolation performed best. The corresponding plots for the best interpolation are shown in Section B.3. Further, we give the percentage of the population that is outperformed by the genetic-algorithm-improved interpolation.

$n_{I}$	1	3	5	7	9	11	13	15
**RMSE** **Population** **Mean**	**2030.11005**	2030.11166	2030.11148	2030.11230	2030.11030	2030.11138	2030.11106	2030.11110
**Lowest** **RMSE in** **population**	**1954.95010**	1954.95013	1954.95016	1954.95013	1954.95005	1954.95009	1954.95020	1954.95015
**RMSE linear** **interpolated**	**2017.79949**	2017.79949	2017.79949	2017.79949	2017.79949	2017.79949	2017.79949	2017.79949
**RMSE spline** **interpolated**	**1971.23755**	1971.23755	1971.23755	1971.23755	1971.23755	1971.23755	1971.23755	1971.23755
**RMSE** **gen. alg.** **improved**	**1907.40084**	1960.21475	1954.94790	1954.94792	1954.95375	1954.97452	1958.57232	1954.97468
**Below** **Best %**	**0.1%**	17.2%	0.1%	0.1%	0.6%	1.01%	14.6%	1.01%
** $n_{I}$ **	**17**	**19**	**21**	**23**	**25**	**27**	**29**	**31**
**RMSE** **Population** **Mean**	2030.11260	2030.11057	2030.11226	2030.11047	2030.11078	2030.11105	2030.11171	2030.11013
**Lowest** **RMSE in** **population**	1954.95010	1954.95007	1954.95011	1954.95014	1954.95007	1954.95010	1954.95003	1954.95021
**RMSE linear** **interpolated**	2017.79949	2017.79949	2017.79949	2017.79949	2017.79949	2017.79949	2017.79949	2017.79949
**RMSE spline** **interpolated**	1971.23755	1971.23755	1971.23755	1971.23755	1971.23755	1971.23755	1971.23755	1971.23755
**RMSE** **gen. alg.** **improved**	1954.97730	1954.99153	1955.00052	1954.99273	1955.02450	1955.02418	1955.01367	1954.98108
**Below** **Best %**	1.3%	1.4%	1.4%	1.4%	1.6%	1.6%	1.4%	1.4%

**Table 4 entropy-24-00718-t004:** Errors for the interpolated data on the Perrin Freres champagne sales data set depending on the number of interpolation points. The errors are shown for the mean interpolation of all populations and for the lowest error in the population, the linear interpolation, the cubic spline interpolation, and the interpolation that was improved using the presented genetic algorithm. We highlighted the interpolation where the genetic-algorithm-based interpolation performed best. The corresponding plots for the best interpolation are shown in Section B.4. Further, we give the percentage of the population that is outperformed by the genetic-algorithm-improved interpolation.

$n_{I}$	1	3	5	7	9	11	13	15
**RMSE** **Population** **Mean**	2320.03501	2320.03532	2320.03403	**2320.03366**	2320.03524	2320.03333	2320.03195	2320.03301
**Lowest** **RMSE in** **population**	2144.04985	2144.05002	2144.04987	**2144.05007**	2144.04991	2144.04986	2144.04981	2144.04995
**RMSE linear** **interpolated**	2264.24606	2264.24606	2264.24606	**2264.24606**	2264.24606	2264.24606	2264.24606	2264.24606
**RMSE spline** **interpolated**	2099.58713	2099.58713	2099.58713	**2099.58713**	2099.58713	2099.58713	2099.58713	2099.58713
**RMSE** **gen. alg.** **improved**	2540.17542	2153.68153	2144.43642	**2144.07624**	2144.16502	2144.15580	2144.14292	2144.16357
**Below** **Best %**	82.5%	6.9%	1.01%	**0.3%**	0.6%	0.6%	0.6%	0.6%
** $n_{I}$ **	**17**	**19**	**21**	**23**	**25**	**27**	**29**	**31**
**RMSE** **Population** **Mean**	2320.03312	2320.03073	2320.03244	2320.03250	2320.03279	2320.03565	2320.03406	2320.03476
**Lowest** **RMSE in** **population**	2144.04987	2144.04986	2144.04967	2144.04982	2144.04997	2144.04985	2144.04976	2144.04999
**RMSE linear** **interpolated**	2264.24606	2264.24606	2264.24606	2264.24606	2264.24606	2264.24606	2264.24606	2264.24606
**RMSE spline** **interpolated**	2099.58713	2099.58713	2099.58713	2099.58713	2099.58713	2099.58713	2099.58713	2099.58713
**RMSE** **gen. alg.** **improved**	2144.09078	2144.13364	2144.17573	2144.13637	2144.16380	2144.10973	2144.10709	2144.13681
**Below** **Best %**	0.5%	0.6%	0.6%	0.6%	0.6%	0.6%	0.6%	0.6%

**Table 5 entropy-24-00718-t005:** Errors for the interpolated data on the monthly airline passengers data set depending on the number of interpolation points. The errors are shown for the mean interpolation of all populations, the linear interpolation, the cubic spline interpolation, as well as for the lowest error in the population and for the interpolation that was improved using the presented genetic algorithm. We highlighted the interpolation where the genetic-algorithm-based interpolation performed best. The corresponding plots for the best interpolation are shown in Section B.5. Further, we give the percentage of the population that is outperformed by the genetic-algorithm-improved interpolation.

$n_{I}$	1	3	5	7	9	11	13	15
**RMSE Population Mean**	19.93996	**19.93999**	19.93841	19.94072	19.93976	19.93873	19.94070	19.93889
**Lowest RMSE in population**	16.55624	**16.55779**	16.55732	16.55753	16.55558	16.55836	16.55719	16.55776
**RMSE linear interpolated**	17.39496	**17.39496**	17.39496	17.39496	17.39496	17.39496	17.39496	17.39496
**RMSE spline interpolated**	18.33872	**18.33872**	18.33872	18.33872	18.33872	18.33872	18.33872	18.33872
**RMSE gen. alg. improved**	18.65257	**16.81653**	16.84539	17.02728	16.84536	16.84545	16.84540	16.84539
**Below Best %**	59.4%	**35.6%**	38.0%	42.20%	38.1%	38.0%	38.1%	38.0%
** $n_{I}$ **	**17**	**19**	**21**	**23**	**25**	**27**	**29**	**31**
**RMSE Population Mean**	19.94029	19.94030	19.93985	19.93939	19.93659	19.94172	19.94023	19.93909
**Lowest RMSE in population**	16.55752	16.55730	16.55810	16.55715	16.55733	16.55603	16.55789	16.55741
**RMSE linear interpolated**	17.39496	17.39496	17.39496	17.39496	17.39496	17.39496	17.39496	17.39496
**RMSE spline interpolated**	18.33872	18.33872	18.33872	18.33872	18.33872	18.33872	18.33872	18.33872
**RMSE gen. alg. improved**	16.84546	16.84545	16.84548	16.84540	16.84535	16.84544	16.84544	16.84546
**Below Best %**	38.1%	38.1%	38.0%	38.1%	38.1%	38.1%	38.2%	38.1%

**Table 6 entropy-24-00718-t006:** Errors for the interpolated data on the monthly mean temperature in Nottingham castle data set depending on the number of interpolation points. The errors are shown for the mean interpolation of all populations, the lowest error in the population, and the interpolation improved using the presented genetic algorithm. We highlighted the interpolation where the genetic-algorithm-based interpolation performed best. The corresponding plots for the best interpolation are shown in Section B.6. Further, we give the percentage of the population that is outperformed by the genetic-algorithm-improved interpolation.

$n_{I}$	1	3	5	7	9	11	13	15
**RMSE Population Mean**	**3.09115**	3.09170	3.09167	3.09055	3.09088	3.09055	3.09166	3.09165
**Lowest RMSE in population**	**2.47879**	2.47858	2.47910	2.47890	2.47886	2.47901	2.47900	2.47875
**RMSE linear interpolated**	**2.61279**	2.61279	2.61279	2.61279	2.61279	2.61279	2.61279	2.61279
**RMSE spline interpolated**	**2.59028**	2.59028	2.59028	2.59028	2.59028	2.59028	2.59028	2.59028
**RMSE gen. alg. improved**	**2.48413**	2.50179	2.50279	2.50406	2.50420	2.50521	2.50512	2.505089
**Below Best %**	**12.6%**	31.3%	32.4%	33.5%	33.8%	34.4%	34.4%	34.1%
** $n_{I}$ **	**17**	**19**	**21**	**23**	**25**	**27**	**29**	**31**
**RMSE Population Mean**	3.09095	3.09177	3.09115	3.09122	3.09146	3.09143	3.09179	3.09023
**Lowest RMSE in population**	2.47887	2.47920	2.47925	2.47899	2.47867	2.47941	2.47885	2.47892
**RMSE linear interpolated**	2.61279	2.61279	2.61279	2.61279	2.61279	2.61279	2.61279	2.61279
**RMSE spline interpolated**	2.59028	2.59028	2.59028	2.59028	2.59028	2.59028	2.59028	2.59028
**RMSE gen. alg. improved**	2.50494	2.50541	2.50529	2.50552	2.50505	2.50547	2.50550	2.50533
**Below Best %**	33.9%	34.6%	34.4%	35%	34.6%	34.6%	34.7%	34.7%

**Table 8 entropy-24-00718-t008:** Errors for the interpolated data on the annual maize yields in Austria data set depending on the number of interpolation points. The errors are shown for the mean interpolation of all populations, the lowest error in the population, and the interpolation improved using the presented genetic algorithm. We highlighted the interpolation where the genetic-algorithm-based interpolation performed best. The corresponding plots for the best interpolation are shown in Section B.8. Further, we give the percentage of the population that is outperformed by the genetic-algorithm-improved interpolation.

$n_{I}$	1	3	5	7	9	11	13	15
**RMSE** **population** **mean**	**9467.34946**	9467.34909	9467.34932	9467.35117	9467.34672	9467.349182	9467.35129	9467.35127
**Lowest** **RMSE in** **population**	**8552.33575**	8552.40507	8552.33623	8552.26190	8552.26676	8552.30054	8552.37868	8552.27861
**RMSE** **linear** **interpolated**	**9641.12086**	9641.12086	9641.12086	9641.12086	9641.12086	9641.12086	9641.12086	9641.12086
**RMSE** **spline** **inteprolated**	**10655.09616**	10,655.09616	10,655.09616	10,655.09616	10,655.09616	10,655.09616	10,655.09616	10,655.09616
**RMSE** **gen. alg.** **improved**	**10204.70770**	10,400.17016	10,404.69095	10,401.94629	10,401.12925	10,401.11260	10,400.83561	10,401.01686
**Below** **Best %**	**96.0%**	96.0%	96.0%	96.0%	96.0%	96.0%	96.0%	96.0%
$n_{I}$	**17**	**19**	**21**	**23**	**25**	**27**	**29**	**31**
**RMSE** **population** **mean**	9467.35137	9467.35207	9467.34967	9467.34998	9467.35055	9467.35120	9467.35351	9467.35240
**Lowest** **RMSE in** **population**	8552.36502	8552.32328	8552.21861	8552.15975	8552.32538	8552.24323	8552.28914	8552.36142
**RMSE** **linear** **interpolated**	9641.12086	9641.12086	9641.12086	9641.12086	9641.12086	9641.12086	9641.12086	9641.12086
**RMSE** **spline** **inteprolated**	10,655.09616	10,655.09616	10,655.09616	10,655.09616	10,655.09616	10,655.09616	10,655.09616	10,655.09616
**RMSE** **gen. alg.** **improved**	10,400.94068	10,401.32128	10,401.59039	10,402.33613	10,401.94342	10,401.55254	10,401.35126	10,401.15196
**Below** **Best %**	96.0%	96.0%	96.0%	96.0%	96.0%	96.0%	96.0%	96.0%

## Data Availability

All data sets previously referenced and discussed in detail.

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
