# Peer review of "Interpolating Strange Attractors via Fractional Brownian Bridges"

_entropy, 2022, doi:10.3390/e24050718_

Round 1

Reviewer 1 Report

    In this paper, the authors proposed a developed method so-called PhaSpaSto-interpolation. The proposed approach was tested and validated with a univariate time series of the Lorenz system, five non-model data sets and tested against a cubic spline interpolation and a linear interpolation. They interpolated the discussed non-model data sets, and show the corresponding improved phase space portraits. The proposed method seems to be useful for interpolating low-sampled time series data sets for, e.g., machine learning, regression analysis, or time series prediction approaches.

In our opinion, these new results are interesting, so we recommend this paper for publication with minor revisions.

Some suggestions:

     1) Introduction needs to be strengthened.
     2) The writing and English language of the whole paper needs to be improved.
     3) Clearly highlight the contribution of your paper.
     4) Tables 1, 2, and 3 need to be adjusted.

Reviewer 2 Report

The paper is original and focused to the signal interpolation using the fractional Brownian bridge (fBB). The most interesting is the application of formulas (8),(9) for the best fractal interpolation design.

Main comments:

  • As seen the penalty L is a stochastic function of the Hurst exponent H. The first question is whether you are able to plot the mean, median and mean value of L as a function of H? Based on the figures, I suppose the optimal H is closed to one.
  • The optimization procedure in the section 3.3.2 is very useful but my suggestion is to called it as Stochastic Optimization Algorithm instead the Genetic Algorithm.
  • The linear interpolation is the unique term, but there are many variants of the spline interpolation. It is necessary to specify it without using the software reference i.e. using exact reference or description.
  • The traditional spline method is a very good referential method which decreases the power of the novel technique of the fractal interpolation. Therefore, I suggest to combine the spline interpolation with additive sequence of fBB's with zero values in the interpolation points. It should be the way how to improve the the spline method.

Minor comments:

  • line 132 the choice of d_E=3 is motivated by the embedded space insight, but Taken's theorem enforced d_E> 2 D_f of the system attractor, when known, of course. please, add the estimate of D_f in the case of Lorenz attractor
  • line 190 the occurence of ?? as missing reference
  • in Fig 3 and previous table, the spline interpolations seems to be the best one method in the case of Lorenz attractor, Isn't it a signal for the method improvement? (see above)
  • The statistical properties of the fBB are also discused in:

Dlask, Martin & Kukal, Jaromír & Sovka, Pavel. (2018). Fractional Brownian Bridge Model for Alzheimer Disease Detection from EEG Signal. 1-4. 10.1109/CSPIS.2018.8642720. 

Dlask M, Kukal J, Poplova ́ M, Sovka P,  Cifra M (2019) Short-time fractal analysis of
biological autoluminescence. PLoS ONE 14(7): e0214427. https://doi.org/10.1371/journal.
pone.0214427
